# Tentative Causes of Brain and Neuropsychological Alterations in Women Victims of Intimate Partner Violence

**DOI:** 10.3390/brainsci14100996

**Published:** 2024-09-30

**Authors:** Julia C. Daugherty, Maripaz García-Navas-Menchero, Carmen Fernández-Fillol, Natalia Hidalgo-Ruzzante, Miguel Pérez-García

**Affiliations:** 1Laboratory of Social and Cognitive Psychology (UCA-LAPSCO), CNRS, University of Clermont Auvergne, 63000 Clermont-Ferrand, France; julia.daugherty@uca.fr; 2Mind, Brain and Behavior Research Center (CIMCYC), University of Granada, 18011 Granada, Spain; carmen.fernandez6@professor.universidadviu.com (C.F.-F.); nhidalgo@ugr.es (N.H.-R.); mperezg@ugr.es (M.P.-G.); 3Department of Health Sciences, Valencian International University, 46002 Valencia, Spain; 4Faculty of Health Sciences, Isabel I University, 09003 Burgos, Spain; 5Department of Developmental and Educational Psychology, University of Granada, 18011 Granada, Spain; 6Department of Personality, Evaluation and Psychological Treatment, University of Granada, 18011 Granada, Spain

**Keywords:** intimate partner violence against women, cognitive, brain injury, neuropsychological functioning

## Abstract

Victims of Intimate Partner Violence Against Women (IPVAW) experience neuropsychological and cerebral changes, which have been linked to several tentative causal mechanisms, including elevated cortisol levels, psychopathological disorders, traumatic brain injury (TBI), hypoxic/ischemic brain damage, and medical conditions related to IPVAW. While these mechanisms and their effects on brain function and neuropsychological health are well-documented in other clinical populations, they manifest with unique characteristics in women affected by IPVAW. Specifically, IPVAW is chronic and repeated in nature, and mechanisms are often cumulative and may interact with other comorbid conditions. Thus, in light of existing literature on neuropsychological alterations in other populations, and recognizing the distinct features in women who experience IPVAW, we propose a new theoretical model—the Neuro-IPVAW model. This framework aims to explain the complex interplay between these mechanisms and their impact on cognitive and brain health in IPVAW victims. We anticipate that this theoretical model will be valuable for enhancing our understanding of neuropsychological and brain changes related to intimate partner violence, identifying research gaps in these mechanisms, and guiding future research directions in this area.

## 1. Neuro-IPVAW: A Theoretical Model for Understanding Mechanisms of Brain and Neuropsychological Alterations in Intimate Partner Violence against Women

Among the extensive number of sequelae and alterations related to intimate partner violence against women [1,2,3], cognitive consequences have been identified, mainly in the domains of attention, memory, and executive functions [4,5,6,7,8]. Cognitive sequelae in the face of intimate partner violence against women (IPVAW) can have dire consequences. For instance, research demonstrates that both physical and psychological violence are related to poorer executive functioning [4], and worse executive functioning is associated with greater difficulties in obtaining resources post-IPVAW [9]. We also know that cognitive deterioration, in domains such as executive functioning, may interfere with advancements in psychotherapy, given that attention and planning are necessary for reaching psychotherapeutic objectives [10,11]. Furthermore, while no studies have examined this directly, some have hypothesized that such cognitive alterations may make it harder to leave the violent relationship [6,12] or increase one’s probability of being revictimized [13].

Despite the relevance of cognitive health in IPVAW victims, very little focus has been given to the potential underlying mechanisms of these alterations. That is, what causes neuropsychological sequelae in victims of IPVAW? Determining the tentative mechanisms for neuropsychological sequelae in this population is unique compared to other clinical populations for several key reasons. First, IPVAW is often chronic and ongoing, resulting in a cumulative health burden. This differs from other clinical populations, such as stroke or traumatic brain injury patients, where neuropsychological and brain alterations typically result from more isolated incidents. Additionally, the diverse manifestations of IPVAW, such as head trauma and sustained psychological abuse, can lead to a wide range of co-existing sequelae. These comorbidities may exert distinct or even synergistic effects, complicating the identification and understanding of underlying neuropsychological mechanisms.

In addition, identifying tentative causal factors for cognitive alterations in this population is challenging due to the lack of longitudinal studies in this area, considering the ethical and practical difficulties of conducting longitudinal research with IPVAW victims. Existing studies are primarily cross-sectional and correlational, often using statistical methods to control for potential confounds. Although the expanding body of literature suggests several tentative mechanisms for cognitive alterations in this population, such as brain injury or PTSD, these studies typically concentrate on a limited number of mechanisms and do not consistently account for other prevalent comorbidities, such as chronic stress and elevated cortisol levels, which may also influence cognitive and brain functioning.

Neuropsychological alterations are the result of structural or functional alterations of the brain [14,15]. Such brain alterations can be tentatively caused by (1) medical conditions, such as neurological illnesses (as is the case with dementia and epilepsy); and/or (2) psychopathology, such as schizophrenia, depression, and post-traumatic stress disorder; and/or (3) adverse environmental conditions, such as poverty or exposure to violence [16]. The relationship between these variables is not unidirectional, such that neuropsychological alterations can invoke symptoms of psychopathology or even increase one’s susceptibility to adverse environmental conditions [17].

Nonetheless, in the case of women victims of IPVAW, the potential mechanisms for neuropsychological alterations stem from their exposure to violence and associated sequelae and injuries. Therefore, when proposing tentative mechanisms for neuropsychological damage in IPVAW victims, these mechanisms must meet three conditions: (1) they must be mechanisms that have been shown in previous empirical literature to cause structural and functional alterations in the brain; (2) there must be scientific evidence demonstrating that these structural and functional alterations cause neuropsychological changes; (3) these identified mechanisms of brain alterations must also occur during IPVAW. These conditions were selected based on both research on IPVAW and established neuropsychological studies of brain and cognitive function. As previous studies have shown, neuropsychological alterations are a direct result of brain structure or function changes [14,15]. Such alterations can arise from a range of sources, including medical conditions, psychopathology, and adverse environmental factors like poverty or violence [16]. Therefore, our first two conditions ensure that any proposed mechanism must have an empirical basis in this neuropsychological framework. Additionally, the third condition was included to ensure that these mechanisms are specifically present in IPV victims, thus grounding the proposal in violence-related sequelae and not arbitrary variables.

Taking these three conditions into account, we propose at least five tentative mechanisms that could explain the neuropsychological alterations found in IPVAW victims: (1) high cortisol levels due to chronic stress exposure; (2) psychopathology(s) related to IPVAW; (3) traumatic brain injury caused by head trauma; (4) anoxic, hypoxic and/or ischemic brain injury caused by strangulation attempts and asphyxia; and (5) damage caused by medical conditions or illnesses related to IPVAW, such as fibromyalgia.

Thus, the present paper seeks to examine these mechanisms of brain and neuropsychological alterations and discusses the potential application of these mechanisms to the case of IPVAW victims. Furthermore, a theoretical model (Neuro-IPVAW) will be proposed to more clearly illustrate tentative mechanisms underlying neuropsychological sequelae and the dynamic relationship between them. To visually represent the relationship between neuropsychological functioning and various forms of IPVAW (including psychological and physical), this review will elaborate on the following figure (Figure 1):

The aim of this article is to propose and discuss a theoretical model outlining five potential mechanisms that may underlie brain and neuropsychological alterations in women victims of IPVAW and to support this model with preliminary empirical evidence. Given the limited research specifically on IPVAW survivors, our theoretical model also draws on neuropsychological principles and empirical evidence from studies on brain and cognitive alterations in other populations, such as those experiencing physical and psychological trauma. For each mechanism, we will present evidence relating it to brain and neuropsychological alterations, the prevalence of these mechanisms in the context of IPVAW, and the existing studies on their cerebral and cognitive impact in IPVAW victims, when they exist.

## 2. Cortisol in the Neuro-IPVAW Model

### 2.1. Cortisol and Brain and Neuropsychological Alterations

Exposure to chronic stress is related to psychopathology and alterations in brain and cognitive functioning [18]. The dysregulation (either an increase or decrease) of endogenous glucocorticoids is the main mechanism through which chronic stress produces these alterations in the brain over the lifespan [19]. Endogenous glucocorticoids (naturally produced cortisol in humans) are a by-product of the hypothalamic–pituitary axis (HPA), which is activated during stress. This stress hormone can pass the blood–brain barrier, where it binds to specific glucocorticoid receptors in the brain [20]. Abnormal increases or decreases in cortisol levels have been found in populations who are subjected to repetitive and chronic stress over time, such as caretakers for Alzheimer patients [21]; boys, girls, and adolescents exposed to repetitive abuse [22]; unaccompanied minor refugees [23]; and individuals exposed to poverty [24]. This relationship has been shown using diverse methodologies for measuring cortisol, including measuring one’s reactivity to stress, daily cortisol levels in saliva, accumulated cortisol in hair samples, or one’s allostatic load [25].

Numerous studies have related endogenous and exogenous cortisol to the structure and functional activity of different brain regions, especially in the medial temporal gyrus, hippocampus, prefrontal cortex, inferior frontal gyrus, and the amygdala [18,26]. Specifically, the acceleration of apoptosis (i.e., programmed cell death) and a reduction in total hippocampus volume have been associated with both circulating cortisol in the bloodstream [27,28] and cortisol measured in hair [29]. With regard to the prefrontal cortex, responsible for higher-order cognitive functions, reduced volume has been related to higher levels of endogenous cortisol [30,31]. Finally, a relationship between cortisol and functional activity in the amygdala has been consistently found, although this relationship is mediated by the type of stimulus/task used (e.g., watching fear-evoking stimuli versus the Montreal Imaging Stress Task) [26]. For a detailed review of the relationship between brain activity and cortisol, please consult Harrewijn et al. [26].

There is also evidence that high cortisol levels can lead to cognitive alterations, especially in the domains of memory and executive functioning. The effect of cortisol on memory during acute stress can be illustrated by an inverted ”U” [32], where low levels of cortisol facilitate memory, but elevated levels impede memorization. Exposure to chronic stress, on the other hand, is shown to have a detrimental effect on memorization [33,34]. High levels of cortisol have also been related to poorer executive functioning [32], although there may be varying effects for different cognitive domains. Specifically, exogenous cortisol has quick-acting detrimental effects on working memory, yet an enhancing impact on inhibition [35].

### 2.2. Cortisol in Women IPVAW Victims

Despite the different approaches used in cortisol research (e.g., saliva versus hair sampling, timing, and study populations), studies consistently show elevated cortisol levels among women who experience IPVAW and who develop post-traumatic stress disorder (PTSD) and/or depression [36,37,38]. Interestingly, though, recent studies have shown that the severity of violence is related to elevated levels of cortisol, after controlling for the effects of depression and PTSD [39,40]. Considering these findings combined, it appears that IPVAW alone, and not uniquely in the presence of psychopathology, is related to higher levels of cortisol. Along these lines, when chronic cortisol levels are measured through hair cortisol, the results show that women victims have higher cortisol levels compared to women who had not experienced such violence [39,41] and that cortisol levels are better explained by IPV severity than by PTSD or depression [39].

### 2.3. Cortisol and Brain Alterations in IPVAW

Despite evidence showing elevated cortisol levels in IPVAW victims and the link between high cortisol and brain alterations in other groups, no studies have explored cortisol’s role in neuropsychological changes in IPVAW victims. Nonetheless, evidence indicates that children exposed to high levels of IPV before the age of 2 are significantly more likely to exhibit prolonged cortisol responses and differences in emotion regulation. These early changes are later followed by cognitive and behavioral differences [42]. Regarding brain alterations, no studies have been conducted to examine the relationship between cortisol levels and the brain alterations experienced by IPVAW victims.

## 3. Psychopathology(s) in the Neuro-IPVAW Model

### 3.1. Psychopathology(s), the Brain, and Neuropsychological Functioning

Various mental health disorders have been closely associated with brain and neuropsychological alterations. Extensive research has documented cerebral changes linked to conditions such as schizophrenia, depression, obsessive–compulsive disorder, anxiety, and PTSD. Consequently, it is unsurprising that a range of neuropsychological alterations corresponding to these psychopathological conditions has also been observed [43].

Among the various psychopathological diagnoses, post-traumatic stress disorder (PTSD), complex PTSD, chronic anxiety, and depression are highly prevalent in women who have experienced IPVAW [3,44]. Moreover, various studies have shown that several of these diagnoses contribute to explaining neuropsychological alterations in IPVAW victims [5,7,12,45]. Thus, here we will focus on PTSD, complex PTSD, anxiety, and depression as possible mechanisms involved in the brain and neuropsychological alterations found in women who have suffered IPVAW.

#### 3.1.1. PTSD

Post-traumatic stress disorder (PTSD) is a disorder that can arise from one’s subjective experience during exposure to a traumatic event, especially in cases where the experience exceeds the individual’s ability to integrate their emotional experience, or the person experiences a threat to his/her life or physical or mental integrity [16,46].

The intrusive symptoms, avoidance of related stimuli, hyperarousal, and negative alterations in mood and cognition are characteristic of this disorder [16]. Numerous investigations have demonstrated alterations in brain function, structure, and biochemistry associated with PTSD. A fear learning and memory brain network, centered in the prefrontal cortex, hippocampus, and amygdala, has generally been described as playing a key role in the pathology of PTSD [47]. Furthermore, it is important to note that changes in the function, structure, and biochemistry of this network appear to underlie the cognitive–affective dysfunction observed in PTSD [47]. The cognitive alterations observed in individuals with PTSD include impairment in attention, executive functions, memory, and psychomotor processing speed (for a review, see Lavoie et al. in Noggle and Dean [48] and Vasterling and Walt [49]).

While different types of traumatic events are related to PTSD onset, the symptoms of this disorder usually appear and worsen when the traumatic event is interpersonal, such as in IPVAW [50]. In IPVAW victims, PTSD has generally been positioned as the most frequent psychopathological disorder. Estimates indicate that between 31% to 84.4% of IPVAW victims develop PTSD, with a weighted average prevalence of 63.8% [51,52]. While studies in this population are scarce, there is emerging evidence of structural and functional neural correlates of PTSD in women who have experienced IPVAW. Specifically, post-trauma neural correlates have been found in the prefrontal cortex [53], occipital cortex [54,55], temporal sulcus [54], limbic system [56,57], anterior cingulate cortex [58,59], and insula [60,61]. It has been hypothesized that the association between these brain regions and PTSD could partly be due to sustained hyperactivation of the emotional and limbic systems in response to trauma experienced in intimate partner relationships [58] and in childhood [56].

Some studies have also examined functional brain changes that occur during threatening and/or violent situations in order to better understand the predictors for the development and chronicity of PTSD. The women victims who participated in these studies had recently experienced episodes of IPVAW or were residing in shelters and safe houses after recently ending or fleeing a violent relationship, thus continuing to live in fear of their partner/ex-partner [62]. One of these functional neuroimaging studies found that, when anticipating adverse stimuli, women experiencing PTSD linked to IPVAW showed greater activation in the right middle insula and bilateral anterior insula compared to women who had not experienced violence from their partners [60]. Additionally, connectivity analysis in this study revealed that changes in activation of the right middle insula and bilateral anterior insula were more strongly associated with amygdala activation changes in the control group than in IPVAW victims with PTSD symptoms. These results combined demonstrate the link between PTSD in the context of IPVAW and brain correlates, most notably in areas concerned with limbic functioning and emotional regulation.

The literature linking cognitive alterations to PTSD symptomology in IPVAW victims is scarce. Nonetheless, to date, several studies have found associations between both subjective and objective neuropsychological functioning and PTSD [5,12,13,54]. One of the first studies in this area compared women survivors of IPVAW with PTSD, without PTSD, and women who had not experienced IPVAW [12]. This study found that the three groups were similar in language ability, with the exception of the raw WAIS-III vocabulary score, which was marginally lower in the subjects with PTSD compared to the non-PTSD and the non-exposed group. This study also found that women with a PTSD diagnosis took significantly longer to complete an executive functioning task (i.e., cognitive set shifting) than unexposed women and those exposed without PTSD, the latter having an intermediate performance between the other groups.

Expanding on this study, Twamley et al. found that, among a sample of IPVAW victims, more severe PTSD symptoms were associated with slower processing speed, and more severe dissociative symptoms were associated with poorer reasoning performance [5]. These authors further suggest that the cognitive slowing observed in PTSD may be attributed to a decrease in attention due to the need to allocate cognitive resources to cope with psychological distress or unpleasant internal experiences resulting from the violence. More recently, research has focused on the link between PTSD and executive functions (EF). A study on self-reported EF found that perceived EF was negatively associated with PTSD symptomology in women victims [54] such that more severe PTSD symptoms were related to poorer perceived EF. Rodríguez-Ipiña and Guzmán-Cortés expanded on these findings by studying the relationship between PTSD symptoms and objective EF performance [13]. Similarly, findings revealed a strong inverse relationship between PTSD symptoms and the total score of executive functions, as well as lower executive functioning scores in women with a PTSD diagnosis versus those without.

Therefore, the present literature provides convincing evidence to support PTSD as a tentative mechanism involved in brain and cognitive alterations among IPVAW victims. However, to improve our understanding of this relationship, it is crucial that future studies include women survivors of IPVAW without PTSD, a control group for other comorbid pathologies, and other mechanisms such as those suggested in the present review [63].

#### 3.1.2. Complex PTSD

Although the literature has largely focused on PTSD, it is essential to also consider complex PTSD (CPTSD). The concept of CPTSD originated from a widespread consensus among a large number of clinical and research professionals who recognized that the existing diagnostic criteria for PTSD did not adequately capture the severe, chronic, and potentially escalating psychological impact experienced by victims of severe interpersonal traumas [64]. Therefore, in the ICD-11 [65], a distinction was established between PTSD and CPTSD based on the manifested symptoms, which are likely related to the type of trauma experienced. Consequently, CPTSD is a distinct diagnosis that includes the three main symptoms of PTSD (trauma re-experiencing, avoidance of traumatic memories, and persistent sense of threat) as well as an additional set of symptoms known as disturbances in self-organization (affective dysregulation, negative self-concept, and disturbances in relationships) [66].

The existing research on CPTSD has been carried out with various populations, particularly in children who have reported repeated abuse (see, for example, Bertó et al. [67], Cloitre et al. [68], and Ford [69]). Regarding IPVAW survivors experiencing CPTSD, to our knowledge, there are only two studies that take into account the most recent ICD-11 criteria applied to this population [70,71]. Both studies indicate that CPTSD is present in this population and warrants investigation, assessment, and treatment. The first study reports high percentages of CPTSD (39.50%) and lower percentages of PTSD (17.90%) among IPVAW victims. It also reveals that certain risk factors, such as fear of the aggressor, are associated with classic PTSD symptoms, while other factors (such as low resilience or maladaptive emotional regulation strategies) are related to the disturbances in the self-organization symptoms that define CPTSD. These findings highlight a new and necessary avenue of research [70].

Unfortunately, no study has examined the neuropsychological correlates of CPTSD according to the contemporary ICD-11 criteria, and brain studies are practically non-existent. One pioneering study compared individuals with CPTSD and PTSD recruited from clinical services, demonstrating distinctive neural processes during threat processing. In this study, Bryant et al. found that participants diagnosed with CPTSD showed greater bilateral activation of the insula and right amygdala compared to participants with PTSD during supraliminal processing of faces expressing fear, anger, and disgust (i.e., threat processing) [72]. Although these findings need to be extensively replicated, they are quite consistent with the symptomatic profile of CPTSD [66] given the role of the insula in the emotional dysregulation symptoms characteristic of CPTSD [72].

Due to the high prevalence of CPTSD in IPVAW victims, the symptomatology characteristics of CPTSD, the preliminary findings of neural correlates, and the lack of literature regarding the associated neuropsychological sequelae, further research is needed to better understand the cognitive and brain underpinnings of CPTSD, particularly in women who have experienced IPVAW.

#### 3.1.3. Generalized Anxiety Disorder

Generalized anxiety disorder (GAD) is an anxiety condition [16] marked by persistent apprehension about improbable events and a pervasive difficulty in controlling excessive worry. Individuals with GAD frequently experience cognitive difficulties, such as trouble concentrating or a “blank mind”, along with self-esteem issues that significantly impair their overall functionality [73].

Anxiety disorders are associated with alterations in neural circuits of fear, such that “bottom-up” processes in the amygdala that react to threat are exaggerated, and the regulation of these processes by the prefrontal cortex and hippocampus are altered [74]. Specifically, structural imaging studies suggest that pediatric patients with GAD have elevated gray matter to white matter ratios in the superior temporal lobe, and they show increased amygdala volume corresponding with stress-induced tension [75]. Functional brain studies of patients with GAD reveal that the limbic circuits, particularly the amygdala, play a crucial role in the fear response. For example, imaging research shows that patients with GAD exhibit heightened activation of both the amygdala and the insula during the processing of negative emotions [76,77]. In response to seeing angry faces, these patients showed that symptom severity was positively related to an elevated right amygdala response [78]. Regarding neuropsychological alterations, studies show that subjects with GAD perform worse than controls in cognitive functions such as selective attention, working memory, inhibitory control, and social cognition [73].

In the context of IPVAW, GAD is notably prevalent among victims [79], with reported prevalence rates ranging from 34% [80,81] to 56.1% [82]. Several studies report a positive association between experiencing IPVAW and increased anxiety levels in women victims of IPVAW, even after taking into account demographic variables such as age, education, and income [83,84]. In another study, abused women exhibited greater severity of anxiety symptoms when the abuse was more frequent, intense, or severe [85]. Furthermore, anxiety problems in these women directly impact their social, work, and personal lives [86].

Despite the lack of evidence at the cerebral level [54], one study has examined the association between anxiety and neuropsychological alterations in women victims of IPVAW. Again, literature examining the relationship between GAD and neuropsychological and brain alterations in IPVAW victims is scarce. Nonetheless, several preliminary studies have demonstrated a relationship between GAD and several cognitive domains, such as memory and attention [87,88]. García-Navarro et al. analyzed differences in memory and attention in terms of mental health and psychopathology (including anxiety, depression, and stress) [87]. The findings indicate that higher anxiety scores are associated with enhanced alternating attention, with increased arousal potentially serving as the underlying mechanism for this relationship. On the contrary, results indicate an inverse relationship between anxiety and working memory, immediate memory, and visual memory, such that increased anxiety was related to poorer performance in these cognitive domains. According to the authors, this inverse relationship could occur because worried thoughts derived from anxiety interfere with the temporary storage of working memory. A subsequent study found that anxiety could exacerbate motor deficits among women who experienced IPV-related traumatic brain injury [88]. These studies reflect a complex interaction between physical injuries and anxiety resulting from violence on cognitive functioning. Therefore, more studies are needed that contemplate an interaction of pathology resulting from violence and brain injury resulting in cognitive alterations.

#### 3.1.4. Depression

Depression is a mood disorder characterized by mood disturbance, feelings of guilt, sadness, hopelessness, anhedonia, and impairment of cognitive functions [16]. Individuals with depression have a profoundly negative view of the world, the self, and the future, and this negative worldview has been linked to negative biases in attention, interpretation, and memory [89].

In fact, there is extensive literature demonstrating that patients experiencing depression present significant brain and neuropsychological alterations [90,91]. Brain alterations seem to be evidenced in structures belonging to the dorsal cortico-limbic networks, and by lower functional connectivity [92,93], specifically in the dorsolateral prefrontal cortex and anterior cingulate cortex. These networks mediate attentional capacity and arousal [94], information processing, and autobiographical memory [95].

Research indicates that neuropsychological alterations may develop as a consequence of brain changes associated with depression [94]. Specifically, patients with depression exhibit significant impairments in executive functions, memory, attention [96,97,98], and processing speed [99] when compared to healthy controls. During the early stages of depression, executive functions, psychomotor speed, and visual memory are particularly affected. While patients in remission often experience improvements in attention tasks, their performance typically does not return to the levels observed in healthy individuals (for a review, see Shenal et al. [93] and Roca et al. [97]).

Depression disorders have a high prevalence among women victims of IPVAW [3,97,100,101,102], with rates between 37.6% and 83%. These rates significantly exceed rates of depression among women in the general population.

Despite the lack of evidence at the cerebral level [54], one study has examined neuropsychological alterations in women victims of IPVAW. Torres-García et al. found low scores in the areas of memory, logical reasoning, visual perception, and attentional control in women victims of IPVAW with depressive symptoms [103]. In addition, the more severe the depressive symptoms, the greater the impairment in short-term memory.

In conclusion, while studies on neuropsychological and brain alterations in IPVAW victims with PTSD, CPTSD, anxiety, or depression are limited, the existing findings align with those observed in individuals who have not experienced intimate partner violence. This could indicate that IPVAW victims present neuropsychological and brain alterations linked to these psychopathologies, which are very prevalent in this population.

## 4. Traumatic Brain Injury in the Neuro-IPVAW Model

### 4.1. TBI and Brain and Neuropsychological Alterations

Other mechanisms potentially causing brain and neuropsychological alterations in IPVAW victims include traumatic brain injury and chronic traumatic encephalopathy. A traumatic brain injury (TBI) is damage to the brain caused by an external force (e.g., head getting hit with an object, head striking a hard surface, rapid acceleration/ deceleration, or blast injuries), resulting in diffuse axonal injury, concussion, contusion, hematoma, skull fractures, and/or cerebral laceration [104,105,106]. The severity of TBI can range from mild to severe, depending on the types of clinical signs and symptoms presented following the injury [106]. In cases of repeated TBI, patients may develop chronic traumatic encephalopathy (CTE), a neurodegenerative condition with an insidious progression that often presents with diverse cognitive issues (e.g., difficulties in thinking, motor disturbances) [105]. As described later, diagnostic criteria for TBI can vary in studies on brain trauma in IPVAW victims. However, for the purposes of this article, we will refer to the new diagnostic criteria for mild TBI presented by The American Congress of Rehabilitation Medicine [106] and the universal definition of TBI: “an alteration in brain function or other evidence of brain pathology caused by an external force such as a blow or injury to the head, severe neck rotation, and acceleration/deceleration” [107].

Despite disagreement about categorical definitions for severity and the upper thresholds for ”mild” TBI [106], traumatic brain injuries generally continue to be defined as either mild or severe. These definitions are expected to evolve over time as empirical evidence from multidimensional biomarkers helps refine and improve the accuracy of diagnoses. Currently, severe TBIs are differentiated from mild TBIs in terms of symptom severity and persistence. A severe TBI is defined by a loss of consciousness lasting more than half an hour or amnesia lasting more than 24 h after the traumatic event. Mild TBIs, while resulting in shorter durations of loss of consciousness and amnesia, must cause at least one or more clinical signs (observable alterations, such as a loss of consciousness) or several clinical symptoms with a laboratory finding or neuroimaging results (please see Silverberg et al. [106] for a more in-depth explanation of diagnostic criteria).

Two events related to trauma can be identified: the primary event and the secondary event [108]. The primary event refers to the immediate trauma caused by an external force associated with the injury (e.g., a blow to the head), usually resulting in neuronal injuries such as shearing of brain tissue or vasculature and neurotransmitter imbalances [109,110]. The secondary event, on the other hand, refers to the reactive processes that occur in the hours or days following the initial event [105]. During this secondary event, progressive changes develop in the brain’s neurochemistry and neurometabolism. These changes can lead to short- and long-term sequelae, including physical, behavioral, cognitive, and emotional problems, resulting in structural changes to neurons, intracranial pressure, a disruption to the blood–brain barrier, hemorrhagic progression of a contusion, metabolic dysfunction, and even atrophy or an increased risk of dementia [105,109,111,112]. Both primary and secondary injuries can lead to functional impairments, affecting cognitive, emotional, social, and sensorimotor domains [113,114]. A wide variety of cognitive sequelae related to TBI have been found, including problems with processing speed, visuomotor coordination, attention, memory, and/or executive function [115]. While the type of cognitive impairment may depend on the force and nature of the impact (e.g., direction, location), severe TBIs can produce neuropsychological alterations in any domain.

As a part of the secondary injury, some individuals may develop post-concussive syndrome (PCS) following a mild TBI, characterized by a constellation of sensory and physical symptoms such as headaches, dizziness, insomnia, tinnitus, sleep disturbances, and attention and memory problems [116]. While approximately 90% of symptoms are transient and will disappear over the course of days or several weeks, those who continue to have symptoms for over three months post-injury are considered to have persistent PCS. Research indicates that patients with mild TBI who develop persistent PCS are more likely to experience enduring cognitive deficits, particularly in memory, learning, and executive functioning [116].

Critically, PCS symptomatology commonly overlaps with symptoms of depression, anxiety, and post-traumatic stress disorder [117]. Therefore, caution is required when making a diagnosis, and psychiatric care is essential when comorbidities are present [116]. Beyond the overlapping of symptoms, it is also important to consider that TBI and comorbidities, such as PTSD, may have synergistic effects with TBI [118,119]. Research suggests that the combined impact of stress and TBI is driven by underlying mechanisms such as inflammation, excitotoxicity, oxidative stress, dysregulation of the hypothalamic–pituitary–adrenal axis, and autonomic nervous system dysfunction [118].

As was alluded to previously, individuals who have experienced more than one TBI are more susceptible to accumulative effects on emotion, behavior, and cognition. They may also develop chronic traumatic encephalopathy (CTE), a neurodegenerative disease that can only be definitively diagnosed after death through a neuropathological examination [120]. CTE is characterized by the abnormal buildup of hyperphosphorylated tau protein (p-tau) in neurons located around blood vessels, particularly at the depths of the cortical sulci [121]. Despite the need for posthumous lab testing for diagnostic accuracy, the scientific literature over the past several decades has elucidated a series of clinical signs and symptoms indicative of CTE, primarily among sports-related injuries in boxing and American football athletes. Such research shows a triad of symptoms in mood, behavior, and cognition, which are oftentimes accompanied by Parkinsonism and ataxia (lack of voluntary movement) [122]. In terms of cognitive impairment, patients commonly report deficits in memory, executive function, attention, language, and visuospatial abilities [122,123]. Patients may also present early motor symptoms, including mild dysarthria and balance impairments, which can subsequently evolve into more severe conditions such as ataxia, coordination difficulties, spasticity, and Parkinsonism [123].

### 4.2. TBI in IPVAW Victims

Regarding the prevalence of TBI among women who have experienced IPVAW, a literature review by Haag et al. [124] reveals that, among empirical studies, the prevalence of TBI ranges from 28.1% among women who reported physical and/or psychological violence [125] to 75% among women who experienced physical violence [6]. In studies that included only women who reported head injuries, 100% of them had experienced a TBI [126,127]. Of these studies, between 30% and 81% of the women had lost consciousness after an act of physical violence [128].

The wide variance in TBI prevalence may be attributed to differences in study methodologies: some studies recruited women based solely on their status as IPVAW victims (including those who experienced only psychological violence), while others focused exclusively on women who had sustained physical injuries. Another difficulty in establishing a more precise prevalence is the varying recruitment locations, such as shelters [6,126,129,130,131], hospital emergency departments [128], courts [132], health centers [133,134], and veteran centers [125,135]. Moreover, it is likely that the published rates do not reflect the reality of all women victims, as IPVAW is extremely underreported [108,136,137]. Indeed, it is estimated that only 17–21% of women who retrospectively screen positive for an IPV-related head injury seek medical treatment [7,137]. Finally, there is no universal consensus on diagnostic criteria for TBI. The most commonly used criterion considers a physical blow to the head, face, or neck, an alteration of consciousness, post-concussion symptoms, or a combination of these factors [124]. Nonetheless, the description of a “physical blow to the head” does not reflect other types of alterations in consciousness (AIC), which help to determine the presence and/or severity of brain injury [106,107]. A scoping literature review revealed that only three studies considered an extensive list of AICs, reflecting IPV-BI rates of 74–100% [138]. As far as we know, only two studies differentiate between a single TBI episode and multiple episodes, showing that 75% of women victims had suffered multiple, and sometimes “too many to count” [6,131].

Despite evidence of chronic TBIs in this population, very little research has been conducted on CTE diagnosis among female victims. The first study linking CTE to IPVAW was published in 1990 [139], where post-mortem analysis of a woman victim showed similar morphological brain alterations to those found in other CTE patients. In the past three years, two subsequent studies have been published on CTE diagnosis in the context of IPV [140,141]. These studies underscore the critical nature of repetitive head trauma in CTE pathogenesis in the context of IPVAW, and they highlight the importance of screening for cognitive and behavioral symptoms in victims who have reported cumulative head injuries.

### 4.3. TBI and Brain and Neuropsychological Alterations in IPVAW Victims

Regarding TBI-related brain alterations found in neuroimaging, only four studies have been conducted with IPVAW victims [6,54,142,143]. These studies indicate both poorer brain connectivity related to TBI [6,142] and structural brain alterations [6,54,142,143]. More specifically, in terms of functional neuroimaging findings, Likitlersuang and colleagues found lower connectivity between the isthmus cingulate seed and several regions, such as the superior parietal and frontal cortices, when comparing 23 IPV victims with TBI and 22 IPV victims without TBI [142]. Valera and Kucyi, on the other hand, gathered neuroimaging data while 22 IPV victims of TBI were at rest (i.e., resting state fMRI) in order to examine brain connectivity with the right anterior insula (rAI), a region that is central to emotional control and decision-making [6]. Findings revealed that low connectivity between the rAI region and the PCC/posterior cingulate precuneus was related to higher brain injury scores. These findings held true even after controlling for IPV severity, age, head motion, and trauma experienced in childhood, as well as psychopathology. Furthermore, Valera and Kucyi found that brain connectivity was positively correlated with cognitive performance in the domains of memory and learning [6]. Thus, these findings combined demonstrate the impact of TBI on brain connectivity in regions that are critical for emotion control and cognition.

In terms of structural alterations, there is evidence of a morphological impact of IPVAW-related TBI on women victims. Along these lines, in a sample of 20 IPVAW-TBI victims, Valera and colleagues demonstrated that greater brain injury severity is related to a lower fractional anisotropy (i.e., the directionality of white matter diffusion) in the posterior corona radiata and the superior corona radiata [143], a region that has shown changes in youth football players post-season [144]. A subsequent study found significant differences between a group of 28 IPV survivors versus 27 women who had not experienced IPVAW in the left superior temporal sulcus, right precuneus, left anterior cingulate cortex, and right subparietal sulcus. Further, they demonstrated that structural differences in these areas were negatively associated with IPVAW-related TBI even after controlling for a range of covariates, such as adverse childhood experiences, psychopathology, and strangulation [54]. More recently, Likitlersuang and colleagues demonstrated that victims of IPVAW with TBI exhibited increased cortical thickness in the right paracentral gyrus compared to individuals with TBI from other non-IPVAW causes [142]. Taken together, these findings underscore the substantial impact of IPVAW-related TBIs on both brain structure and function in women victims, revealing significant morphological and connectivity alterations that persist even after accounting for various confounding factors.

With regard to brain alterations in CTE, forensic autopsies reveal pathophysiological findings in women who sustained repetitive head injuries from their partner [140,141]. These findings were consistent with diagnostic criteria for hyperphosphorylated tau (p-tau) in neurons surrounding blood vessels at the level of the cortical sulci [121]. More specifically, tau pathologies were detected in the inferior parietal lobule, the ventrolateral frontal cortex, and the anterior temporal lobes, as well as the dorsolateral frontal cortex [140,141]. Importantly, the authors also identified low levels of age-related tau astrogliopathy, indicating that the observed tau abnormalities could not be attributed to normal brain atrophy associated with aging [141]. These studies mark the very first documented cases of CTE pathology in women victims of IPVAW. As the authors note, despite the extensive literature documenting CTE pathology in males, there is a glaring lack of research on CTE in women [140], particularly in victims of IPVAW. Given the high probability of repetitive head injuries in victims of physical IPVAW, estimated to be around 50% [6,131], research in this area is urgently needed.

Physical violence has been associated with multiple neuropsychological alterations in domains such as attention and concentration, visuoconstructive abilities, motor processing speed, and fluency in IPVAW victims [5,6,12,129,131]. Only a few studies have empirically investigated the relationship between TBI and neuropsychological alterations in IPVAW victims, specifically focusing on the sequelae related to chronic TBIs [6,131]. These studies found that the number of TBIs was related to poorer performance in verbal memory, cognitive flexibility, and learning. These findings may account for the memory, attention, and concentration issues frequently reported by IPVAW victims, emphasizing the importance of cognitive screening when TBI is detected.

## 5. Hypoxic and/or Ischemic Brain Injury in the Neuro-IPVAW Model

### 5.1. Hypoxic and/or Ischemic BI and Brain and Neuropsychological Alterations

Hypoxic brain injuries arise when the brain’s oxygen supply is insufficient, but blood flow to the brain may still be adequate. Oxygen in the brain is derived from the blood; thus, its delivery relies on proper cerebral blood flow and adequate oxygen concentration in the bloodstream, which is influenced by intake through breathing. Consequently, hypoxic brain injury can result from disruptions in cerebral blood flow, systemic issues that reduce the blood’s oxygen content, or obstruction of the airways (e.g., due to asphyxia or smothering) [145]. Ischemic brain injury, on the other hand, is a type of acute brain injury that results from impaired blood flow to the brain, causing deficiencies in oxygen and nutrient delivery [146].

Strangulation can induce both hypoxic and ischemic brain injuries by simultaneously limiting blood flow and oxygen delivery to the brain. Strangulation is defined as external pressure on the vascular and/or respiratory pathways of the neck, preventing the flow of air and blood to the head [147,148]. Strangulation can reduce cerebral blood flow in several ways. Firstly, it can result in anoxia or hypoxia by blocking oxygen to the brain via the carotid arteries (ischemia) and through the trachea and larynx in the airways (asphyxia). Additionally, occlusion of the carotid arteries can trigger the carotid sinus reflex, leading to arrhythmia or cardiac arrest, and consequently, less blood flow to the brain (hypoxic–ischemic). Secondly, attempts at strangulation can increase intracranial pressure by blocking blood outflow through the jugular veins. The restriction of cerebral blood flow can cause neuronal death and, in severe cases, the death of the victim, even without any external signs. To determine if strangulation has resulted in an acquired brain injury, it is crucial to gather information on any alterations in consciousness [149].

While research on the cerebral and neuropsychological consequences of non-fatal strangulation is limited, there is an emerging body of literature on sexual choking and its neuropsychological consequences [150]. This body of literature is important to understanding the specific impact of non-fatal strangulation on the brain, as studies conducted on sexual choking assess individuals who did not experience concurrent traumatic brain injury. This is particularly important due to the fact that incidences of IPVAW-related strangulation and TBI are essentially indistinguishable. Sexual choking, which disproportionately affects women, has been related to a series of short-term health sequelae including alterations of consciousness (such as headaches and a loss of consciousness) as well as long-term sequelae for neurological health (such as strokes, seizures, and vascular abnormalities) [150]. In terms of brain alterations, recent neuroimaging studies have revealed that frequent exposure to sexual strangulation is linked to distinct neural activation patterns [151,152]. Specifically, one study identified increased activation in several brain regions in individuals with a history of sexual strangulation compared to those without such a history during verbal and visual working memory tasks [152]. Notably, this study excluded participants with a traumatic brain injury within the past year or more than two lifetime traumatic brain injuries. These results suggest that sexual strangulation may impact the allocation of neural resources under conditions of increasing cognitive load [152], and that strangulation may lead to distinct neural functioning independent of TBI.

Beyond sexual choking, broader research on hypoxic–ischemic brain injury—caused by factors such as cardiopulmonary arrest, respiratory failure, and carbon monoxide poisoning—has also found that brain alterations can occur post-injury, potentially affecting any neuropsychological domain [153]. A literature review on cognitive sequelae linked to hypoxic–ischemic brain injury noted that the most common consequences are found in the domains of attention, processing speed, memory, and executive function [153]. Due to the limited research on the sequelae of acquired brain injury resulting from hypoxia, patients are often given the same rehabilitation programs used for traumatic brain injury. However, it is important to exercise caution when applying clinical guidelines designed for TBI to hypoxic–ischemic injuries. Emerging empirical evidence suggests that cognitive outcomes for hypoxic–ischemic injuries may differ—and potentially be worse—compared to those resulting from TBI [154].

### 5.2. Hypoxic and/or Ischemic BI in IPVAW

Professionals working with IPVAW victims increasingly acknowledge strangulation as a critical form of violence due to its significant impact on both short- and long-term health, as well as its association with severe violence and homicide [155,156,157,158,159,160]. Existing studies indicate a high prevalence of IPVAW-related strangulation attempts [157,160,161,162,163,164,165], with 50–68% of women victims having experienced at least one instance of strangulation [7,165] and 82% of these women experiencing it more than once [157]. The sole study investigating the incidence of strangulation with alterations in consciousness (AICs, an indicator of underlying brain alterations) found that 27% of women who experienced physical IPVAW reported at least one strangulation-related AIC, while 12% experienced multiple AICs related to strangulation [149].

Alterations resulting from strangulation attempts often go unnoticed because they frequently occur alongside other forms of violence [157]. This is not surprising, as strangulation is linked to heightened levels of violence and even homicide [157,165,166]. In fact, 97% of women who have experienced non-fatal strangulation report being struck by various means [167]. Additionally, the confusion and post-concussion symptoms resulting from traumatic brain injury can make it challenging for IPV victims to differentiate between symptoms caused by blows and those from strangulation attempts. Strangulation attempts may also go unnoticed due to the fact that up to 50% of strangulation cases present no visible signs [160]. In fact, many victims of strangulation die each year without any apparent external signs of neck injury [162,163,167].

Strangulation attempts in IPVAW victims are associated with various short- and long-term health concerns. A literature review on strangulation in intimate partner violence describes immediate problems (amnesia, ataxia, dizziness) and delayed issues (strokes) [168]. Consequently, the brain and neuropsychological alterations that IPVAW victims experience may be related to both immediate brain damage from lack of oxygen and cerebral blood flow as well as delayed damage caused by strokes and neuronal death.

### 5.3. Hypoxic and/or Ischemic BI and Brain and Neuropsychological Alterations in IPVAW Victims

In examining the cerebral correlates of strangulation, neuroimaging studies of IPVAW survivors often classify alterations in consciousness (AICs) resulting from strangulation-induced anoxia or hypoxia as traumatic brain injuries. This approach aligns with standard clinical practice, which recognizes that the resulting symptoms are often similar. Additionally, many women experience incidents in which they are strangled and their head is hit simultaneously [6], making it hard to compare “groups”. Nonetheless, given recent neuroimaging research that demonstrates distinct neural activation patterns for women experiencing frequent strangulation [151,152], more research is needed to understand the specific cerebral consequences of hypoxic and/or ischemic brain injury as a consequence of IPVAW-related strangulation.

Only one study to date has studied the relationship between strangulation attempts and brain structure while controlling for the potential effect of TBI in a sample of IPVAW victims [54]. In this study, authors examined structural brain differences between a group of women IPVAW victims and women who had not experienced IPVAW. Findings revealed that having experienced strangulation by a partner was negatively associated with the cortical thickness of the left horizontal branch. This relationship held even after controlling for the impact of adverse childhood experiences, TBI, PTSD, and depression. While this study did not test for strangulation-related AICs (i.e., strangulation was included as a yes/no dichotomous variable), this finding suggests that the cerebral consequences of strangulation might be distinct from those stemming from traumatic brain injury resulting from head impacts. Valera and colleagues provide some support for this hypothesis. Their study found that the correlation between TBI and structural brain changes diminished when accounting for the effects of strangulation in IPVAW victims [143]. While this does not establish a direct causal relationship, it implies that strangulation could account for a portion of the observed variability in brain alterations associated with IPVAW.

In conclusion, the combined and emerging neuroimaging research on the neural correlates of strangulation (both in and out of the IPVAW context) tentatively point towards a specific impact of hypoxic and/or ischemic injury on the brain.

To our knowledge, only a handful of studies have systematically investigated the relationship between neuropsychological performance and strangulation in IPVAW victims [149,169]. The majority of studies predominantly rely on symptom checklists, which often conflate psychological symptoms with cognitive deficits (e.g., traumatic immobility due to fear versus agnosia/lack of initiation) [168]. Nonetheless, this literature consistently demonstrates that amnesia is a significant concern among non-fatal strangulation survivors (up to 54%) [170,171].

Among studies that have used objective cognitive testing, Valera and colleagues found that a higher number of strangulation attempts correlated with poorer performance in verbal memory—specifically, the ability to learn and recall a list of words—even after controlling for the impact of TBIs sustained during the abusive relationship [149]. In other words, similar to TBIs from blows, an accumulative dose–response effect is observed, where symptoms worsen with an increasing number of strangulation attempts [172]. Therefore, the alterations caused by oxygen restriction to the brain in IPVAW victims may be more severe than in other clinical populations, as women often experience multiple instances of strangulation over the course of an abusive relationship [149].

In another study, Muir and colleagues completed two neuropsychological follow-up evaluations (at 1 week and at 3 months) with eight women who had attended the emergency room following a strangulation attempt [169]. Findings demonstrated cognitive impairment had not only persisted, but had actually worsened, particularly in the areas of cognitive flexibility and inhibitory control. These findings underscore the critical need for neuropsychological evaluations in cases of strangulation, as they highlight the distinct and enduring impact of hypoxic and ischemic insults to the brain, particularly in instances of repeated exposure.

## 6. Medical Conditions in the Neuro-IPVAW Model

The DSM-5-TR [16] and various neuropsychological manuals [173,174] have identified numerous health conditions that neuropsychologists must consider due to their significant impact on cognitive, behavioral, and emotional well-being, including but not limited to cardiovascular diseases, diabetes, sexually transmitted infections (e.g., HIV), genetic and cellular disorders (e.g., cancer), and immunological conditions (such as fibromyalgia and lupus).

Research has found that cardiovascular diseases, such as hypertension and heart failure, can cause cognitive impairment [175,176]. These impairments are notably observed in the areas of information processing, executive dysfunction, attention, and immediate memory deficits [174] due to reduced blood flow and low cerebral oxygenation. Diabetes has been linked to an increased risk of cognitive impairment, affecting areas such as spatial and episodic memory, processing speed, and executive functions [177]. This is due to chronic hyperglycemia and vascular complications, which lead to changes in the brain’s white matter, among other effects. Autoimmune diseases have also been related to neuropsychological alterations in verbal and visual short- and long-term memory due to underlying inflammation and neuronal damage [178]. Fibromyalgia is linked to memory, attention, planning, inhibitory control, and information processing problems [179,180] derived from prolonged activation of the sympathetic nervous system and neurochemical changes. Cancer and its treatments, such as chemotherapy [181], can negatively impact cognitive functions such as verbal memory, attention, psychomotor speed, and executive function. Sexually transmitted diseases, such as human immunodeficiency virus (HIV), can cause direct neuronal damage and neuroinflammation, resulting in HIV-associated neurocognitive disorder [182], characterized by problems in attention, learning and memory (working and episodic), psychomotor impairment, and executive dysfunction [174,183]. For an in-depth review of these medical conditions, please see Armstrong and Morrow [173], and Parsons and Hammeke [174].

It is important to note that these medical conditions are highly prevalent in women who have experienced IPVAW [44,86,184,185,186,187]. In fact, the prevalence of health problems among IPVAW victims is two to three times higher than among women who have not experienced any violence [188].

Prevalence studies have shown that women who have suffered IPVAW are more likely to suffer from cancer [189]. Specifically, there is a 1.47 to 4.28 times greater risk of cervical cancer in IPVAW victims than in those who have not been exposed [186,190]. The chronic stress experienced during a violent relationship increases the risk of developing cardiovascular diseases, such as hypertension or stroke, by 33% in IPVAW victims [185,191,192]. There is also a high prevalence of chronic pain [193], fibromyalgia [194], and autoimmune diseases such as lupus [195] among IPVAW victims—70–95% higher than among women who have not experienced violence [185]. Other studies indicate a 46% increased likelihood of developing diabetes [185,196,197] and other physical and sexual health problems, such as sexually transmitted infections [198]. Specifically, IPVAW victims are 2 times more at risk than women who have not experienced this type of violence [186].

However, despite the fact that women victims of IPVAW suffer from these medical conditions more frequently, there are no studies exploring how these conditions impact their neuropsychological and brain functioning.

Additionally, victims may experience other illnesses that, while not directly affecting cognition, generate high levels of stress with neuropsychological consequences. For instance, gastrointestinal disorders like irritable bowel syndrome and dermatological conditions such as psoriasis can be exacerbated by chronic stress, thereby increasing the emotional and cognitive burden [186,199,200]. Consequently, the cumulative impact of violence and multiple chronic health conditions can lead to more severe cognitive impairment and a reduced quality of life.

## 7. Why the Mechanisms of Neuropsychological Impairment in IPVAW Are Specific: Comorbidity(s), Interactions, and Repetitive Occurrence

In our view, findings from previous research on brain and neuropsychological alterations related to chronic stress, psychopathology, traumatic brain injury (TBI), strangulation, and medical conditions in other populations cannot be directly extrapolated to IPVAW victims for two primary reasons: (1) the accumulation and/or combination of causal mechanisms leading to brain and neuropsychological damage experienced by IPVAW victims, who may experience TBI, strangulation, chronic stress, PTSD, CPTSD, anxiety, and/or depression simultaneously; (2) the repetitive and sustained nature over time of the traumas, TBIs, and strangulation attempts experienced by IPVAW victims.

Regarding the first specific feature, it is important to recognize that the mechanisms causing brain and neuropsychological alterations in women victims are notably complex and unique. For these women, the brain may be simultaneously exposed to elevated cortisol levels, TBI, hypoxia or ischemic brain injury, and psychological conditions such as PTSD, CPTSD, anxiety, and depression, in addition to the effects of chronic illnesses like fibromyalgia.

The literature provides limited evidence on the brain and neuropsychological consequences of simultaneous exposure to multiple mechanisms such as elevated cortisol, TBI, anoxia or hypoxia, PTSD, CPTSD, anxiety, depression, and chronic conditions like fibromyalgia. Few studies have investigated these combined effects, but existing research—primarily focusing on veterans with both PTSD and mild traumatic brain injury (mTBI)—indicates that individuals with both conditions exhibit more severe neuropsychological alterations and distinct brain patterns compared to those with only one of the conditions (PTSD or mTBI) [201,202]. More recently, specific alterations in the connectome have also been found in the fronto-limbic system in veterans suffering from PTSD + mTBI [203]. To the best of our knowledge, there are no studies on brain alterations in people who have suffered TBI + strangulation, PTSD + strangulation, PTSD + strangulation + TBI, etc.

In the case of female survivors, the probability of these mechanisms occurring simultaneously in the same person is high according to the prevalences indicated in the previous sections. However, to our knowledge, there are no studies on the cumulative effect of these mechanisms in female survivors.

In addition to the first unique characteristic, it is crucial to consider the repetitive and sustained nature of the traumas, TBIs, and strangulation attempts experienced by IPVAW victims over time. Regarding repeated brain injuries, it is crucial to note that the sequelae following one or more brain injuries are potentially different in IPVAW victims compared to other populations [7]. Existing literature in populations such as athletes [204,205], military personnel [206,207], and individuals experiencing TBI from accidents [208] has limited applicability, as they primarily focus on male, young, relatively healthy individuals without comorbid psychopathological disorders and with different types of traumas (e.g., impact or blast vs. strangulation). However, brain injuries in IPVAW victims may present unique characteristics that influence their impact on the brain and cognitive functioning.

The differences between TBIs in IPVAW victims and other populations can be delineated in three aspects. Firstly, existing studies indicate that brain trauma and subsequent outcomes are different between men and women [209,210,211]. Studies have shown that female athletes, after experiencing sports-related trauma, exhibit greater deterioration in reaction time and more severe post-concussion symptoms, both objectively and subjectively, compared to their male counterparts, even when accounting for helmet use. Additionally, women athletes were 1.7 times more likely than males to experience cognitive impairments [209]. In military populations, women veterans report more somatosensory and vestibular problems within the first 30 months post-trauma and experience more nonspecific symptoms such as headache, depression, fatigue, appetite changes, and sleep disorders [212]. Despite findings from studies involving athletes and veterans, research indicates that the manifestation of TBIs in women is distinct and necessitates targeted investigations. These studies should avoid confounding sex biases to accurately understand the sequelae experienced by women following trauma.

Lastly, the literature indicates that TBIs caused by violence have worse consequences compared to TBIs caused by other means [213]. Intentional TBIs, compared to those caused by accidents, appear to have poorer recovery for daily tasks that rely on cognitive abilities such as memory, comprehension, and communication. In relation to military populations, while many also suffer from psychiatric problems [207], the types of blast injuries from explosions differ from the head impacts often sustained by women victims (often repeated over years). A systematic review comparing sequelae related to blast versus impact trauma indicates that there may be distinct cognitive impairments between the two types of traumas [214]. This evidence suggests that sequelae caused by blast-type trauma are not representative of those experienced by IPVAW victims. Therefore, studying neuropsychological alterations specifically related to TBIs caused by IPVAW is merited.

When considering emotionally traumatic events, it is important to emphasize their repetitive nature and the fact that they are inflicted by someone who is emotionally connected to the victim. This has led to the development of a new nosological entity called complex post-traumatic stress disorder (CPTSD), which extends the symptomatology of classic PTSD [64]. Unfortunately, to date, neuropsychological and brain studies have primarily focused on women diagnosed with PTSD, with limited research addressing CPTSD.

## 8. Future Directions on Mechanisms of Brain and Neuropsychological Alterations

The Neuro-IPVAW model, a theoretical framework based on neuropsychological principles and recent empirical evidence, presents several tentative mechanisms for explaining brain and neuropsychological alterations in IPVAW victims. Nevertheless, none of these mechanisms have been systematically examined in IPVAW victims, and most of the proposed relationships are based on a very reduced number of studies or studies with small samples. For example, research has established a relationship between psychopathology and neuropsychological alterations [5,12,215]. In female survivors of IPVAW, psychopathological conditions can help explain their neuropsychological deficits [5,7]. Conversely, neuropsychological sequelae may also serve as risk factors for the development of psychopathology, as demonstrated in other populations. For instance, a longitudinal cohort study found that children with atypical executive function development during early life were at a higher risk of developing PTSD following trauma exposure later in life [216]. Understanding the mechanisms underlying these alterations in women who have experienced IPVAW can provide valuable insights into their complexity and offer potential strategies for preventing the development of these consequences.

Furthermore, as mentioned, the tentative mechanisms proposed in the Neuro-IPVAW model do not necessarily occur in isolation; rather, several mechanisms may occur simultaneously. For example, women who have reported strangulation have also reported several repetitive head injuries [143]. In addition, brain and neuropsychological alterations may be associated with the interaction of repeated TBI, strangulations, traumatic events, and chronic stress associated with IPVAW. Future research should investigate whether the cumulative or combined effects of multiple mechanisms lead to more severe consequences than the individual impacts of each mechanism considered separately. This approach could reveal interactions between mechanisms that amplify their overall effects, providing a more comprehensive understanding of the complex interplay involved.

On the other hand, conducting longitudinal studies is essential to establish a cause-and-effect relationship between IPVAW, the interaction of various mechanisms, and the development of cerebral and neuropsychological consequences. However, implementing such studies with IPVAW victims presents significant challenges. To conduct this research, it would be necessary to initially evaluate girls or young women who have not experienced IPVAW and track them over time to determine how many report IPVAW, develop neuropsychological and brain sequelae, and how these are associated with the mechanisms under investigation. This has serious ethical implications. If a longitudinal study of this nature were conducted, once an IPVAW situation was identified, the victim would need to be placed in a position of complete safety. Consequently, the research might not allow sufficient time to observe the resulting sequelae, their association with the severity and duration of the violence, or the relationship with the potential causal mechanisms identified.

Furthermore, the proposed tentative mechanisms in the Neuro-IPVAW model are non-exhaustive. Future research should study if other mechanisms could also be linked to the brain and neuropsychological sequelae in this population. For example, we highlight consumption of alcohol and psychotropic drugs or exposure to poverty. Regarding the former, previous studies conducted with North American women who have experienced IPVAW revealed a high rate of alcohol abuse and dependence [52]. In other populations, alcohol consumption has been related to the appearance of brain and neuropsychological alterations [217,218]. However, when studying these potential mechanisms in IPVAW, it is important to take into account cultural differences in substance use and alcohol consumption. For example, a recent study conducted in Spain showed Spanish IPVAW victims may turn to psychotropic medication rather than alcohol to cope with their symptoms [219]. In addition, they related this consumption to psychopathology and found that psychopathological symptoms were not related to alcohol consumption, whereas post-traumatic symptomatology was related to psychotropic drug consumption. In this regard, a possible additional mechanism that could explain the cerebral and neuropsychological sequelae in women who have experienced IPVAW is the consumption of psychotropic drugs. Thus, research in this regard should be conducted, and cultural factors must also be considered when analyzing these issues.

Another potential, yet unexplored, factor is exposure to poverty, food deprivation, and limited access to resources. These conditions affect a significant number of women who have experienced IPVAW [220,221]. Research in other populations suggests that such socioeconomic stressors can impact cognitive functioning and brain health [222]. However, the effects of poverty on women who have experienced IPVAW, particularly regarding potential brain and neuropsychological alterations, have not been adequately investigated. Thus, the Neuro-IPVAW model is likely to expand the number of tentative causes that explain the brain and cognitive consequences of IPVAW in women in the future.

In addition to the need for further research on the mechanisms underlying brain and neuropsychological alterations in IPVAW victims, significant gaps remain in understanding the neuropsychological and cerebral sequelae specific to women. First, the incidence and prevalence of these alterations are not well established. While some neuroimaging and neuropsychological studies have identified impairments in small sample groups, they do not provide a clear picture of the broader prevalence among IPVAW victims compared to the general population. Population-based studies are necessary to determine the extent to which these sequelae affect women who have experienced IPVAW, offering a clearer understanding of the full impact on brain and cognitive functions.

On the other hand, another important line to explore is the functional consequences of brain and neuropsychological sequelae in female victims of IPVAW. Some studies have suggested that they have important implications for social, occupational, and emotional regulation [223,224]. For example, executive functioning problems in these women have been shown to be associated with their ability to obtain resources one year after experiencing IPVAW, regardless of the women’s income level [9]. Moreover, issues with response inhibition or decision-making resulting from violence could, among other factors, complicate a victim’s ability to leave an abusive relationship [6,12]. However, in contrast to the important implications of these sequelae, the number of studies is still insufficient.

It is also crucial to examine how neuropsychological impairments evolve following abuse and to assess the effectiveness of neuropsychological rehabilitation in addressing these impairments. When brain damage occurs and neuropsychological changes manifest, the brain initiates several repair mechanisms, including spontaneous recovery and brain plasticity, to restore its functions [225]. Even without formal rehabilitation, some recovery of neuropsychological functions can occur after acute brain damage, influenced by factors such as the severity of the damage and the patient’s age [226]. However, most existing studies on recovery post-brain damage focus on patients with singular conditions, such as one traumatic brain injury or cerebrovascular diseases. In contrast, IPVAW victims often experience multiple forms of trauma, such as TBI combined with hypoxia or PTSD. Thus, it is important to investigate whether the patterns of brain and neuropsychological recovery observed in other patient populations also apply to women who have suffered IPVAW. We need to determine whether recovery occurs in these individuals, if it varies across different neuropsychological functions, and how it might differ based on the underlying mechanisms or their combinations. Additionally, it is essential to evaluate the effectiveness of neuropsychological rehabilitation for IPVAW victims and to understand how its efficacy might be influenced by the mechanisms of trauma, the severity of the violence, and other related factors.

Finally, there is a notable lack of research into the forensic implications of neuropsychological alterations in women who are victims of IPVAW, despite some preliminary studies on this topic [227,228]. Investigating these neuropsychological and brain changes is crucial, as it could significantly impact economic compensation for sequelae experienced by survivors after leaving the perpetrator, as well as the credibility of the testimony and their imputability in criminal proceedings [229]. Comprehensive study of these sequelae will provide essential evidence to develop and validate forensic assessment tools tailored for women who have experienced IPVAW and to integrate such assessments systematically into judicial processes involving survivors.

In conclusion, several critical avenues for future research in this domain warrant exploration. First, it is imperative to systematically investigate the tentative mechanisms proposed by the Neuro-IPVAW model in studies involving women survivors. This approach will allow for a nuanced understanding of how these mechanisms impact brain and cognitive functioning. Additionally, it would be helpful to examine the interactions between these mechanisms, their contributions to both short- and long-term cognitive and cerebral sequelae, and their functional implications for daily living and forensic contexts. Beyond the five tentative mechanisms identified in the Neuro-IPVAW model, researchers may also consider other potential contributing factors, including socioeconomic and cultural variables, which may mediate these relationships. To facilitate a unified international approach and ensure consistency in research findings, adopting standardized measures across studies is essential. Noteworthy efforts in this regard include the ENIGMA consortium, which works to harmonize data across research sites, and the BELIEVE project, which has developed a free multi-lingual neuropsychological assessment battery for IPVAW survivors [230].

While additional empirical evidence for the Neuro-IPVAW model is necessary, the theoretical framework proposed in this paper offers valuable guidance for practitioners and front-line workers in identifying key red flags for potential brain and cognitive alterations. As noted, cognitive impairments in memory and executive functioning can significantly impact an individual’s ability to access resources and progress in therapy. Thus, observing signs of the five tentative mechanisms should alert practitioners to potential impacts on cognition, and thus intervention efficacy. If detected, practitioners can then tailor their services to address these needs. For example, if brain injury and resultant memory impairment are identified, psychologists or social workers might integrate compensatory cognitive training strategies, such as external memory aids, to enhance therapy adherence and efficacy.

Moreover, engaging front-line workers and practitioners in research is crucial. These professionals provide invaluable insights into the functional impacts, symptom presentations, and barriers to seeking help related to brain and cognitive injuries. Their perspectives and access to critical data can significantly advance both research and practical interventions. For instance, academic researchers and medical institutions are collaborating to collect data on biomarkers for IPVAW-related brain injuries and tracking medical visits to distinguish between long-term sequelae resulting from hypoxic versus traumatic brain injuries.

We encourage both front-line workers and researchers to engage with associations such as Pink Concussions, which serve as highly beneficial resources by connecting practitioners, stakeholders, and researchers to enhance research, awareness, and outreach. Establishing international and cross-disciplinary coalitions is essential for deepening our understanding of brain and neuropsychological alterations in this population and improving overall care.

## 9. Conclusions

In summary, IPVAW victims experience neuropsychological and brain alterations that could be tentatively caused by several mechanisms, including sustained levels of cortisol, psychopathological alterations, TBI, hypoxic/ischemic brain injury, and medical illness related to IPVAW.

Although these mechanisms of brain alterations and resulting neuropsychological damage are well-documented in other clinical populations, they exhibit unique characteristics in the context of IPVAW victims: (1) the accumulation and/or combination of causal mechanisms leading to brain and neuropsychological damage, where alterations in cortisol, PTSD, CPTSD, anxiety, depression, TBI, and hypoxic/ischemic brain injury may be experienced simultaneously; (2) the repetitive and sustained nature over time of trauma, TBIs, and strangulation attempts.

Drawing from the existing literature on neuropsychological alterations in other populations and acknowledging the unique context of women experiencing IPVAW, we present a model to illustrate the intricate relationships among these mechanisms—the Neuro-IPVAW model. We expect that this model could be useful (1) to better understand the neuropsychological and brain alterations related to intimate partner violence against women; (2) to identify gaps in the study of these mechanisms; and (3) to guide future research about these mechanisms.

## Figures and Tables

**Figure 1 brainsci-14-00996-f001:**
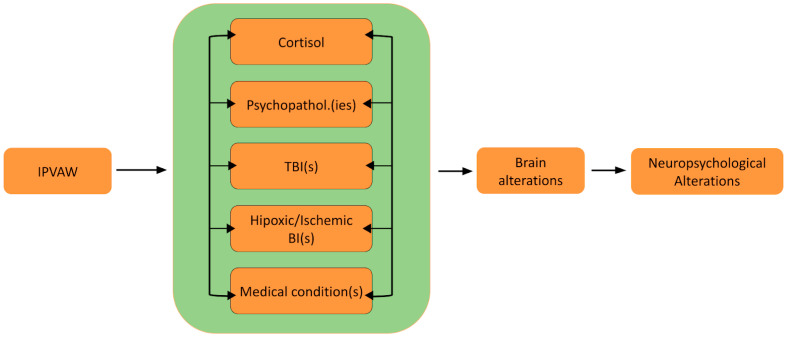
The Neuro-IPVAW model: Tentative mechanisms of brain and neuropsychological alterations in intimate partner violence against women. Note. BI: Brain injury; IPVAW: Intimate Partner Violence Against Women; TBI: Traumatic brain injury.

## Data Availability

No new data were created or analyzed in this study. Data sharing is not applicable to this article.

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
