# Peer review of "Tentative Causes of Brain and Neuropsychological Alterations in Women Victims of Intimate Partner Violence"

_brainsci, 2024, doi:10.3390/brainsci14100996_

Round 1

Reviewer 1 Report

Comments and Suggestions for Authors

Dear Editor,

I have attempted to review the article and have identified the following issues:

Line 83. A reference should be provided for the three conditions mentioned. Line 89. What is meant by "damage caused by medical conditions related to IPVAW, such as fibromyalgia"? Figure 1. On what basis did the authors make their assumptions? Have they conducted previous research? Could they explain better how they arrived at this theoretical construct? Line 152. “Recent studies have shown that the severity of violence is related to elevated levels of cortisol” – the relationship seems to be rather inverted, meaning that cortisol levels are related to the severity of violence. … The rest of the article is a hodgepodge of studies, many of which are inconclusive for women victims of IPV or identify alterations in other populations, such as children exposed to domestic violence or in experimental studies. The article is extremely heavy to read as it describes various neuropsychological disorders or alterations that may result from violence, while it would be much more interesting to streamline the review, including only what is strictly relevant to the investigated topic. In this version, it seems more like a book chapter than a review for a journal.

Reviewer 2 Report

Comments and Suggestions for Authors

The authors present a comprehensive paper on brain injuries and neuropsychological consequences due to Intimate Partner Violence where they review the literature and propose a new model for this population-the Neuro-IPVAW model. Overall, they have done an excellent job on supporting this area via literature review. They include the areas of hypoxic, ischemic brain injury, PTSD, biological aspects (cortisol) and mental health. The final section indicates the importance of moving forward to study all areas not just the brain injury. This paper will make a great contribution to the field of brain injuries. 

Reviewer 3 Report

Comments and Suggestions for Authors

The authors had undertaken an extensive and systematic review of research at the intersection of IPV, mental health, and TBI. As a scholar who works at this intersection, I find the paper to be compelling, accessible, and comprehensive. This paper will not only make an important contribution to the literature, but will be an important resource for others seeking to conduct research at this intersection. My only recommendation is to encourage the authors to add some recommendations to their conclusion. As they consider both future research and interventions, what would they propose scholars and practitioners pay attention to so that they can further address the gaps in the literature identified here and serve survivors more effectively and efficiently?
